


# Measurement of the ice-nucleating particle concentration using a mobile filter-based sampler on-board of a fixed-wing uncrewed aerial vehicle during the Pallas Cloud Experiment 2022

Alexander Böhmländer[1], Larissa Lacher[1], Kristina Höhler[1], David Brus[2],
Konstantinos-Matthaios Doulgeris[2], Jessica Girdwood[3,4], Thomas Leisner[2], and Ottmar Möhler[1]

[1]Institute of Meteorology and Climate Research, Karlsruhe Institute of Technology, Karlsruhe, Germany
[2]Atmospheric Composition Research, Finnish Meteorological Institute, Helsinki, Finland
[3]Centre for Atmospheric and Climate Physics, University of Hertfordshire, Hatfield, Hertfordshire, AL10 9AB, UK
[4]now at: National Centre for Atmospheric Science, School of Earth, Atmospheric and Environmental Sciences, University of Manchester, Manchester, M13 9PL, UK

**Correspondence:** Ottmar Möhler (ottmar.moehler@kit.edu)

**Abstract.** A novel filter-based sampler was deployed during the Pallas Cloud Experiment (PaCE) 2022 for a one-month period in September and October 2022 in Finnish Lapland around 5 km north of the Sammaltunturi station. This area frequently features low-level clouds during autumn. The sampler was deployed on-board of an uncrewed aerial vehicle (UAV) and on the ground. Two filters were deployed simultaneously on the ground and on the UAV to enable a comparison between the two vertical levels. The dataset contains 14 INP spectra that feature a temporal overlap at both altitudes, a handling blank filter to assess possible contamination during handling and one additional sample from both setups without the temporal overlap. The dataset is the first of its kind, providing altitude-based INP concentrations in Finnish Lapland. There is no clear systematic difference between INP concentrations measured at the different altitudes. The INP concentration is variable over the period measured and also does show some differences on the vertical level. The connection to synoptic conditions and ambient measurements might provide a better understanding of the origin, lifetime, and distribution of INPs in Finnish Lapland.

## 1 Introduction

Ice-nucleating particles (INPs) induce the primary ice formation of liquid pure water droplets at supercooled conditions above about -38 °C (e.g., Koop et al., 2000). Mixed-phase clouds (MPCs) exist in the temperature range -38 to 0 °C, where the fraction of ice inside the cloud is controlled by the presence of INP and affects its properties, such as lifetime, radiative budget and precipitation. Precipitation events, and by that the lifetime of a cloud, are linked to the presence of an ice phase in clouds, especially at higher latitudes (e.g., Field and Heymsfield, 2015; Mülmenstädt et al., 2015; Heymsfield et al., 2020). The different radiative properties of MPCs have been investigated in relation to their phase in the literature (e.g., Bellouin et al., 2020; Storelvmo, 2017; Shupe and Intrieri, 2004). The nature and sources of atmospheric INPs are understudied, especially with a vertical resolution (e.g., Schmale et al., 2021). The vast majority of INP measurements are performed on ground-based stations (e.g., DeMott et al., 2010, 2017; Kanji et al., 2017; Schneider et al., 2020; He et al., 2021). Linking those

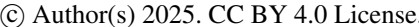


measurements to upper atmospheric INP concentrations is complicated, since aircraft measurements are expensive and are limited in their altitudes (Shupe et al., 2005; Schmale et al., 2021). In the same way, remote sensing techniques to study aerosol-cloud interactions have to rely on models to estimate certain variables such as the INP concentration (e.g., Dietel et al., 2024). Small and lightweight uncrewed aerial vehicles (UAVs) offer a flexible and cheap method to investigate the lower atmosphere, the reachable vertical extent mostly regulated by power considerations (e.g., Altstädter et al., 2018; Lampert et al., 2020; Marinou et al., 2019; Villa et al., 2016; Yu et al., 2017; Schrod et al., 2017; Bieber et al., 2020; Böhmländer et al., 2024a). This is especially relevant for the Arctic and sub-Arctic regions, where the boundary layer is very shallow and low-level clouds are common (e.g., Shupe et al., 2011; Gierens et al., 2020; Dietel et al., 2024).

This report describes filter-based measurements of atmospheric INP concentrations using a simple and lightweight aerosol sampler technique co-located on the ground and on-board of a fixed-wing uncrewed aerial vehicle (UAV). The sampler consists of a filter holder, a mass flow meter and a small and lightweight multi-diaphragm pump. The flow is monitored during operation to ensure constant operation and detection of failures during flight. The co-location offers the simultaneous INP sampling at the ground and during UAV operation, which enables direct comparison at two different altitudes. The technical description of the setup is detailed in Böhmländer et al. (2024a).

## 2 Observation site

The here described measurements have been done as part of the Pallas Cloud Experiment 2022 (PaCE-2022). The sampling location was around 5 km north of the Sammaltunturi station, which is part of the Pallas Atmosphere-Ecosystem Supersite in Finnish Lapland, hosted by the Finnish Meteorological Institute (FMI) (Asmi et al., 2021; Brus et al., 2024) and part of Global Atmosphere Watch (GAW), Integrated Carbon Observation System (ICOS), European Monitoring and Evaluation Programme (EMEP) and the Aerosol, Clouds and Trace Gases Research Infrastructure (ACTRIS). The local vegetation consists of low vascular plants, lichen and moss (e.g., Lohila et al., 2015), while the surrounding forest mainly consists of pine, spruce and birch trees (e.g., Komppula et al., 2005). The anthropogenic impact on the aerosols at the observation site is minor, since it is located inside the Pallas-Yllästunturi National Park and far away from larger settlements (Lohila et al., 2015). The ground setup was located on top of a small hut on an open-field, which was used as a starting and landing area for the UAV. The field is located next to a street with a very low amount of irregular traffic (see also Brus et al., 2024). The goal was to measure at the same time at both altitudes. The altitude for the flight was designated to be just below cloud base to determine the INP concentration close to the cloud.

## 3 Instrument operation

The filters are placed into the filter holding at a clean working environment, wearing gloves and handling the filters itself only with pre-cleaned forceps. Two filter holders are loaded with a filter each and then sealed with closed off black tubing and stored until deployment inside zip-lock plastic bags. The general filter handling is described in detail in Böhmländer et al. (2024a).

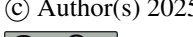



All filters were subjected to an active air flow at a constant altitude, during ascend and descent the pump was turned off. For the PaCE-2022 campaign some filters were flown twice, i.e. after a flight, the filter was not switched with a fresh and clean filter, but the same filter was flown a second time under the same conditions. This leads to an enhanced INP concentration
sensitivity due to the increase in sampled air-volume. All filters were flown on-board of the Skywalker fixed-wing UAV. The collected filters were stored at room temperature at the site (< 4 weeks) and shipped to KIT, where the filters were stored at -18 °C until analysis with the freezing assay Ice Nucleation Spectrometer of the Karlsruhe Institute of Technology (INSEKT). This instrument consists of an actively cooled aluminium block, which can house two 96-well polymerase chain reaction (PCR) plates. The 192 wells of the two plates were filled with Nanopure water and Nanopure water-based suspensions of the
sampled aerosols. The aluminium block houses eight temperature sensors and a camera is located above the sample to detect the brightness of each filled well. The aluminium block and thus the samples in the PCR plates are cooled down at a rate of 0.33 K until all aliquots are frozen. INSEKT is described in detail by Böhmländer et al. (2024a) and references therein.

## 4   Data evaluation and quality control

The raw data produced by INSEKT contains the data of the eight temperature sensors and the grey scale value of each well
derived from the camera output at a frequency of 1 Hz. The freezing temperature is determined by calculating the mean of the temperature sensors as specified in the py_raw_insekt software. The uncertainty of the nucleation temperature is calculated as the standard deviation of the mean, considering a normal distribution. The time when the well freezes is detected by a rapid decrease in the grey scale value. From the amount of frozen droplets in the wells and the total amount of wells filled with the same sample, a frozen fraction is calculated for each sample. Figure 1 shows the frozen fraction of an aerosol sample washed of
a loaded filter in comparison to washing water of a handling blank filter taken during the campaign. The uncertainty associated with the frozen fraction is calculated using the normal approximation of the binomial distribution published by Agresti and Coull (1998) assuming a confidence interval of 95 % (see also Hill et al., 2016; Schneider et al., 2020; Böhmländer et al., 2024a). Using the equations established by Vali (1971) the INP concentration per standard litre of sampled air is calculated and shown in Figure 2 for the two different dilutions. Finally, in Figure 3 the information on the different dilutions is removed and
a single homogenized dataset is shown. This data is given alongside its corresponding frozen fraction for each sample in the dataset presented here. The data is checked considering three potential issues during the analysis with INSEKT, considering three distinctive positions on the frozen fraction scale (0.25, 0.5 and 0.75):

1. Quality of the Nanopure water background: the difference between the frozen fraction of the Nanopure water background and the frozen fraction of the given sample should be smaller than 1 K. If this condition is not met, an error flag is
associated with the data.

2. Separation of suspensions with different aerosol concentrations from the same filter washing water: the difference between the frozen fraction of the suspensions should be smaller than 1 K. If this condition is not met, a warning flag is associated with the data.


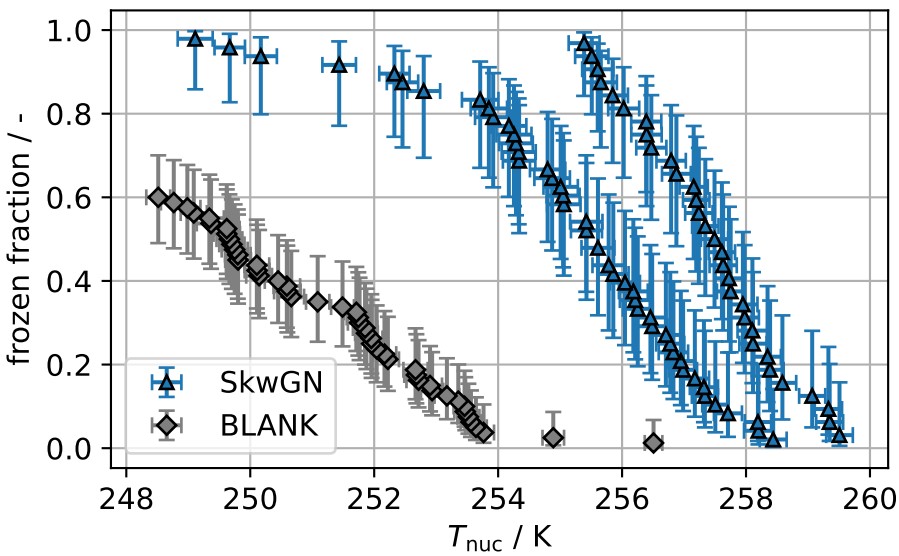

**Figure 1.** The frozen fraction as a function of the observed freezing temperature $T_{\mathrm{nuc}}$ for aerosol particles washed off a ground filter (SkwGN) and off a handling blank filter (BLANK). The two samples from the ground filter are two suspensions with different aerosol concentrations. In this case the left most sample is diluted with dilution factor of 5.

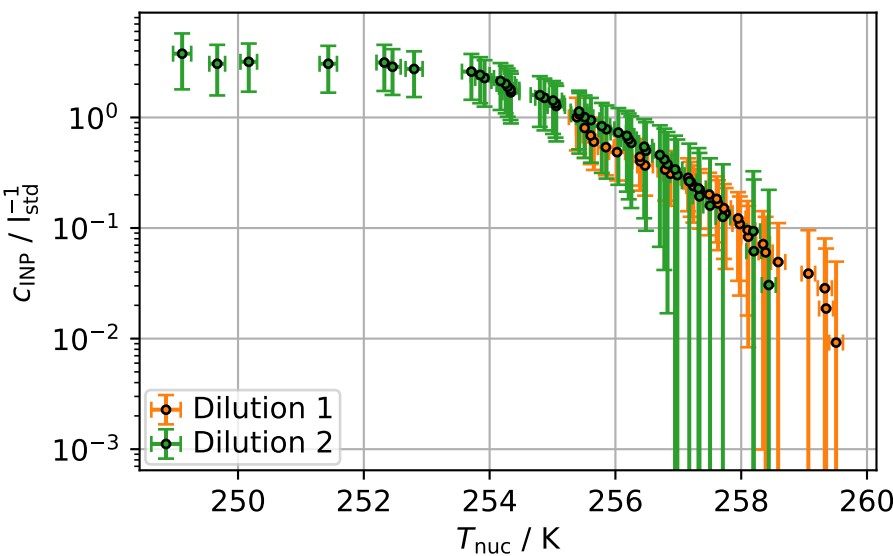

**Figure 2.** The INP concentration as a function of the freezing temperature $T_{\mathrm{nuc}}$. The data is shown for the two suspensions shown in Figure 1.

3. Freezing order: the suspensions should freeze in order, with the one with highest aerosol concentration freezing first. If
   this condition is not met, an error flag is associated with the data.




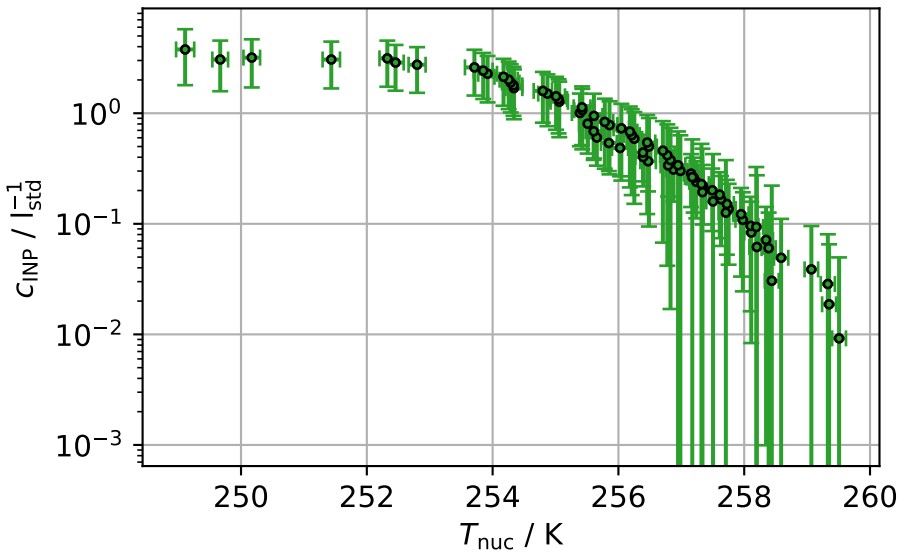

**Figure 3.** The INP concentration as a function of the freezing temperature $T_{\mathrm{nuc}}$ as a homogenized dataset. The two suspensions separately shown in Figure 2 are combined as one for the here presented datasets.

Data with error flags are removed from the datasets. Data with warning flags are manually inspected and removed if necessary.

## 5 Overview of dataset

The datasets are given as netCDF files following the CF-1.11 metadata conventions. There are three types of datasets provided, differing in their sampling condition. One type of dataset is derived from the aerosol washed of a filter loaded on-board of the UAV (Skw), the other type is from an identical setup on the ground (SkwGN). The third data type is for the handling blank (BLANK), which does not contain any data on the INP concentration, but only on the frozen fraction. Two handling blanks were collected during the campaign, but the data from one of the experiments was corrupted and could not be repeated. The dataset contains pairs of UAV and ground filter samples, which have been sampled during the same time period. Figure 4 shows the comparison between the INP concentration at the ground and at an altitude of $200\,\mathrm{m}$ above ground level (agl) sampled on 2022-10-08 09:30:00+0000. The highest INP concentration measured on the ground during this campaign was $13.18^{+6.91}_{-6.53}\,\mathrm{l}^{-1}_{\mathrm{std}}$ ($T_{\mathrm{nuc}} = 252.74\,\mathrm{K}$), while the highest INP concentration on the UAV was $13.41^{+6.70}_{-6.26}\,\mathrm{l}^{-1}_{\mathrm{std}}$ ($T_{\mathrm{nuc}} = 249.75\,\mathrm{K}$). The INP concentrations have been measured between 246.92 (247.61) K and 265.38 (266.58) K on the UAV (ground), limited at the lower temperatures by the Nanopure water background and at the higher temperatures by the INP sensitivity of INSEKT. In total, 14 filter samples from UAV flights are available and can be combined with 14 filter samples taken on the ground with a temporal overlap. One additional ground sample and UAV sample are available, but do not have a temporal overlap. The handling blank filter was taken during the campaign and shows the extent of contamination during the handling of the filters.

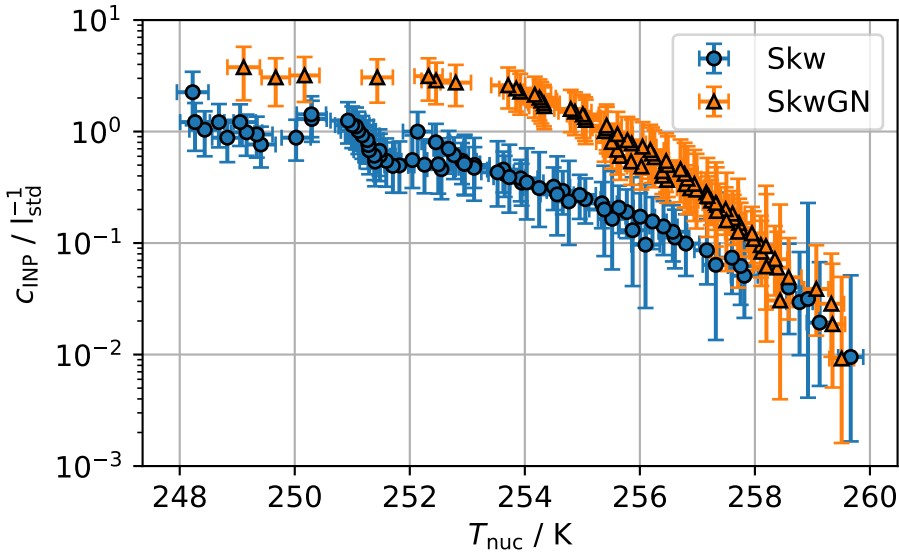

**Figure 4.** The INP concentration as a function of the observed freezing temperature $T_{\mathrm{nuc}}$ for a sample taken on the ground (SkwGN) and on-board of the UAV (Skw).

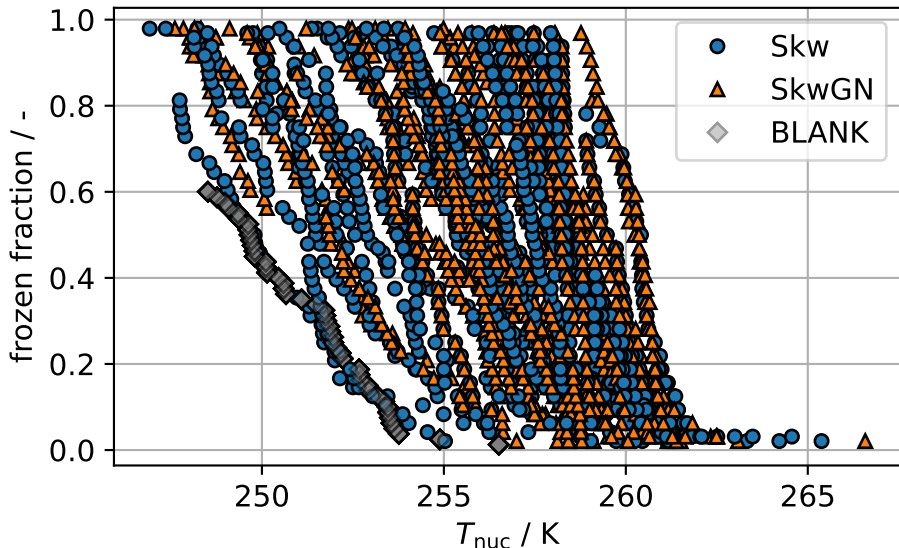

**Figure 5.** The frozen fraction as a function of the observed freezing temperature $T_{\mathrm{nuc}}$ for all filters, split into UAV (Skw), ground (SkwGN) and handling blank (BLANK) samples. Only one sample shows a similar frozen fraction as the handling blank.

The frozen fraction of all filters as well as of the handling blank is shown in Figure 5. Only one sample shows an overlap with the frozen fraction of the handling blank. The dataset is still included, but should be removed for a future analysis.



## 6 Conclusions

The dataset presented provides the first INP concentration measurements using a mobile filter-based setup utilizing a UAV. The data can be used to assert the INP concentration in the vertical column connecting it to different synoptic conditions. Looking at individual cloud cases, especially when multiple samples were taken on the same date, offers also a temporal resolution. The Sammaltunturi station, located just 5 km south of the ground measurements, can be used as a reference for other relevant meteorological variables as well as the measurement of the INP concentration with a high temporal resolution utilizing the Portable Ice Nucleation Experiment (PINE, see also Böhmländer et al., 2025). Since the data is given based on freezing events, the differential spectra can be calculated, obtaining characteristic nucleation temperatures for the aerosol sampled.

## 7 Code and data availability

Datasets are archived under individual DOI at the Zenodo Open Science data archive (https://doi.org/10.5281/zenodo.13911633, last access: 11102024, Böhmländer et al. (2024b)), where a dedicated community Pallas Cloud Experiment - PaCE2022 has been established (https://zenodo.org/communities/pace2022/, last access: 11102024). This community houses the data files along with additional metadata on the datasets. The py_raw_insekt software is available on a public gitlab instance under https://codebase.helmholtz.cloud/insekt/py_raw_insekt.

*Author contributions.* AB did the data analysis and wrote the manuscript. LL, JG and AB performed the flights during the PaCE-2022 campaign. KH reviewed the original manuscript and provided helpful commentary in later stages. TL developed the LabVIEW software to control and interact with the INSEKT. DB and KD prepared and organized the PaCE-2022 campaign. All authors contributed to the proof reading and discussion of the dataset.

*Competing interests.* The contact author has declared that none of the authors has any competing interestes.

*Acknowledgements.* This research has been supported by the ACTRIS IMP GA 871115, the ACTRIS-Finland funding through the Ministry of Transport and Communications, and the Atmosphere and Climate Competence Center Flagship funding by the Research Council of Finland (Grants 337552).
The KIT project contribution was supported by the Helmholtz Association through the research program "Changing Earth - Sustaining our Future". The authors would like to thank the technical team at the Sammaltunturi station for their support during the campaign, and the INSEKT team at KIT for continuous support in developing and operating INSEKT.



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
