# Peer review of "Measurement of the ice-nucleating particle concentration using a mobile filter-based sampler on-board of a fixed-wing uncrewed aerial vehicle during the Pallas Cloud Experiment 2022"

_Earth System Science Data, 2025_

## Author Comment (AC1)

**Author response to RC1**

August 8, 2025

We are grateful for the helpful comments and suggestions from the reviewer. Below the reviewers comments are in blue with our responses in black directly below.

The manuscript presents a unique dataset on ice-nucleating particle (INP) concentrations obtained using a UAV-based mobile filter sampler during the Pallas Cloud Experiment 2022. The dataset is particularly valuable as INP data at different altitudes, especially at the level of low-level clouds, remain sparse, making this dataset a useful contribution for process understanding, comparison with other regions or datasets, or model validation. The methods and materials are well described. Not all details are provided in this publication but can be found in the referenced complementary work (Böhmländer et al., 2024a). The dataset is accessible, complete, and of high quality, with error estimates and uncertainties properly addressed. Some clarifications regarding background subtraction, dilution handling, and data representation would improve usability, but overall, the dataset and data descriptor is well-structured and appropriate for publication.

Some minor points of clarification to enhance transparency and reusability include:
* * *
**1 General comments**

Line 46: Consider specifying the altitude range or average altitude of the UAV measurements.

We have added the range of altitudes in parentheses to the sentence.

old The altitude for the flight was designated to be just below cloud base to determine the INP concentration close to the cloud.

new The altitude for the flight was designated to be just below cloud base to determine the INP concentration close to the cloud. The dataset contains data from the UAV between 405 m and 906 m above mean sea level, resulting in a maximum altitude of 598 m above ground level (agl).

This also had a minor follow-up change by already defining the abbrevation of "above ground level" earlier than before in line 94:

old Figure 4 shows the comparison between the INP concentration at the ground and at an altitude of 200 m above ground level (agl) sampled on 2022-10-08 09:30:00+0000.

new Figure 4 shows the comparison between the INP concentration at the ground and at an altitude of 200 m agl sampled on 2022-10-08 09:30:00+0000.
* * *
Line 52: Consider providing the flow rate used during sampling.

The air flow depends on the altitude of the measurement, therefore we have added a table in the appendix specifying the total amount of air which is calculated from the flight time $t$ and the mass flow $F$

$$V = tF \tag{1}$$

for each filter.

Line 73: The INP concentrations in the data files are reported in m$^{-3}$ but shown in std l-1 in the manuscript; while the unit information is provided in the netcdf file, adding a unit reference in the metadata or a table with units would improve clarity. Also, could you clarify whether any correction was applied for the water background or handling blank?

We have added an additional metadata attribute for the INP concentration in the data files under the description key: Unit is referenced to a standard flow rate at 273.15 K and 101 325 Pa.

The provided data is corrected for the water background, but no correction has been applied originating from the handling blank. We have added a sentence to the last paragraph in the section "Overview of dataset":

old

new The frozen fraction of the handling blank is not subtracted from the frozen fraction of the filter suspensions provided.

Lines 74-75: It appears that dilution information is simply removed rather than averaged, and that you typically used two dilutions - could you confirm if this is correct?

We typically used two suspension during analysis: (1) the undiluted washing water of the filter, and (2) the diluted washing water of the filter with a dilution factor of $d = 5$.

Yes this is correct, since we are providing the data for each freezing event, therefore there essentially is no temperature, where two freezing events of the undiluted suspension and the diluted suspension overlap. This averaging is more relevant, when using a fixed temperature grid, i.e. providing the frozen fraction in 0.5 K steps. Raw events offer a better insight into individual filters, while a fixed temperature grid is more relevant when comparing different datasets via statistical means.

Line 90: When plotting the handling blank data, two fraction frozen curves appear, while only one is shown in Figure 5 – could you explain this discrepancy?

We are thankful to the reviewer for carefully checking our data and finding an issue. We have checked the data again and found an issue in the data analysis of the handling blank filter. We updated the py_raw_insekt software to now treat handling blank filters correctly. The handling blank itself is also analyzed with a water background, therefore the frozen fraction data contains two frozen fraction curves, denoted as "Dilution 0" and "Dilution 1". We provide both in the published dataset, but have not added the required metadata to the netCDF file. We added a new data variable "dilution" denoting the different suspensions and updated the attributes as following:

```
df['dilution'].attrs = {
            'long_name': ('name of the suspension analysed'),
            'short_name': 'dilution',
            'description': ('name of the suspension analysed: nanopure denotes the '
                            'nanopure water background, while Dilution X denotes '
                            'the different suspensions (X=1) and dilutions (X>1).')
            }
```

This is available as an updated version of the described dataset. Due to the update to the py_raw_insekt software, the frozen fraction now also extends to 1.0.

Some INP spectra contain only a few data points, likely due to the described data quality filtering but some further clarification could be beneficial.

We have carefully checked the data again and found an issue in our conversion script. Some filters were included in the data set, although they should have been removed since they did not satisfy the quality control measures. We have therefore updated the dataset and also changed the corresponding doi.

old Datasets are archived under individual DOI at the Zenodo Open Science data archive (`https://doi.org/10.5281/zenodo.13911633`, last access: 11102024, Böhmländer et al. 2024)

new  Datasets are archived under individual DOI at the Zenodo Open Science data archive (`https://doi.org/10.5281/zenodo.16752129`, last access: 06082025, Böhmländer et al. 2024),

The raw data was also reanalyzed with a new version of the py_raw_insekt software, which now automatically calculates also the 95 % confidence interval and therefore the metadata of the NetCDF files now also reflects this change. To accomodate those changes, as well as the changes to the handling blank data via including the dilution as a new variable in the dataset, we have also updated Figures 1 and 5 of the paper and updated the figure description of Figure 5:

old  The frozen fraction as a function of the observed freezing temperature $T_{\mathrm{nuc}}$ for all filters, split into UAV (Skw), ground (SkwGN) and handling blank (BLANK) samples. Only one sample shows a similar frozen fraction as the handling blank.

new  The frozen fraction as a function of the observed freezing temperature $T_{\mathrm{nuc}}$ for all filters, split into UAV (Skw), ground (SkwGN) and handling blank (BLANK) samples. Only some diluted samples show a similar frozen fraction as the handling blank. Note that the handling blank suspension is very close to its Nanopure water background.

[Figure]

Figure 1: The frozen fraction as a function of the observed freezing temperature $T_{\mathrm{nuc}}$ for aerosol particles washed off a ground filter (SkwGN) and off a handling blank filter (BLANK). The two samples from the ground filter are two suspensions with different aerosol concentrations. In this case the left most sample is diluted with dilution factor of 5.

[Figure]

Figure 5: The frozen fraction as a function of the observed freezing temperature $T_{\mathrm{nuc}}$ for all filters, split into UAV (Skw), ground (SkwGN) and handling blank (BLANK) samples. Only some diluted samples show a similar frozen fraction as the handling blank. Note that the handling blank suspension is very close to its Nanopure water background.

**2  Additional minor changes**

We have changed a word to correctly refer to a filter holder in line 49:

old  The filters are placed into the filter holding at a clean working environment, wearing gloves and handling the filters itself only with pre-cleaned forceps.

new  The filters are placed into the filter holder at a clean working environment, wearing gloves and handling the filters itself only with pre-cleaned forceps.

We have updated the reference in the netCDF file to the technical paper:

old  Boehmlaender et al. 2024, in preparation.

new  Boehmlaender et al. 2025, Meas. Tech. Discuss. [preprint], https://doi.org/10.5194/amt-2024-120, in review, 2025.

**References**

Böhmländer, A. J., L. Lacher, and O. Möhler (2024). "Data from filter-based sampler from the ground and on-board of an uncrewed aerial vehicle during the Pallas Cloud Experiment 2022 [data set]". In: DOI: `10.5281/zenodo.13911633`.

---

## Author Comment (AC2)

**Author response to RC2**

August 8, 2025

We are grateful for the helpful comments and suggestions from the reviewer. Below the reviewers comments are in blue with our responses in black directly below.

The manuscript presents a novel dataset on ice-nucleating particle (INP) concentrations obtained during the Pallas Cloud Experiment (PaCE) 2022 in Finnish Lapland.

The study provides valuable altitude-based INP concentration data, and the dataset uniqueness contributes to understanding the INP variability in the region and model evaluation.

The study is scientifically relevant, and the methodology is well described and designed. The data are open, of high quality, and supported by a careful uncertainties assessment.

Some minor clarifications may improve the quality of the manuscript, but it is absolutely appropriate for publication.

The main points of clarification include:
* * *
**1 General comments**

Line 5: Although the meaning of INP is clear throughout the manuscript, consider ti specify the acronym also in the abstract.

We have updated the abstract to include the specification of INPs. We also updated the amount of filters that have an overlapping temporal range. Previously, the data flagged with an ERROR were included as well due to an error in the code (see also comments to reviewer 1). We also addded a table in the Appendix of the manuscript to provide some more information on the samples included in this dataset.

| | |
|---|---|
| old | The dataset contains 14 INP spectra that feature a temporal overlap at both altitudes, a handling blank filter to assess possible contamination during handling and one additional sample from both setups without the temporal overlap. |
| new | The dataset contains 9 ice-nucleating particle (INP) concentration spectra that feature a temporal overlap at both altitudes, a handling blank filter to assess possible contamination during handling and additional samples from both setups without the temporal overlap. |
* * *
Lines 8-9: The abstract could be improved including some numerical values, also as range, mostly when mentions the variability of INP concentrations in time and altitude.

We have added some numerical values to the abstract regarding the INP concentration at $253\,\mathrm{K}$.

| | |
|---|---|
| old | |
| new | The INP concentration at $253\,\mathrm{K}$ varies between $0.15\,\mathrm{l_{std}^{-1}}$ and $3.06\,\mathrm{l_{std}^{-1}}$ on the ground, and between $0.48\,\mathrm{l_{std}^{-1}}$ and $1.69\,\mathrm{l_{std}^{-1}}$ at higher altitudes. |
* * *
Lines 18-19: The sentence is a bit unclear, mostly the last part. Vertical resolution may be misleading, consider to reword this part.

We have adjusted this sentence to be more clear about our intention:

old The nature and sources of atmospheric INPs are understudied, especially with a vertical resolution (e.g., Schmale et al. 2021).

new The nature and sources of atmospheric INPs are understudies. This is especially true for the vertical distribution of INPs in the lower atmosphere (e.g., Schmale et al. 2021).
* * *
Lines 36-37: Despite, Finnish Lapland is well-known some additional information could be extrmely valuable, for instance a small map of the area with the observation site. Or otherwise a short sentence that may help readers to indentify the location (e.g. coordinates).

We have added some additional information and also provided a map, that describes the location in more detail.

old

new The Sammaltunturi station is located at 67°58'24" N, 24°60'58" E, while the measurements with the UAV were conducted above an open space (68°1'10" N, 24°8'52" E), indicated in fig. 1.

[Figure]

Figure 1: Location of Pallas (lower right) and Sammaltunturi (left). The red dot marks the location of the open space used for the UAV operation during PaCE-2022. Figure adapted from Hatakka et al. 2003.

Lines 46: Which is the altitude of the UAV measurements?

The designated altitude of the UAV measurements was not constant during the campaign. The individual altitudes are provided in the metadata of the NetCDF files. We have also added a sentence on the range of altitudes measured, which was suggested by reviewer 1.

old The altitude for the flight was designated to be just below cloud base to determine the INP concentration close to the cloud.

new The altitude for the flight was designated to be just below cloud base to determine the INP concentration close to the cloud. The dataset contains data from the UAV between 405 m and 906 m above mean sea level, resulting in a maximum altitude of 598 m above ground level (agl).

Lines 74-75: What do you mean for "homogenized dataset"? Consider to better clarify what you mean, being "homogenized" rather misleading.

With homogenized dataset we mean a dataset that has the information on the different dilutions removed. We have adjusted the wording to explain it more clearly.

old Finally, in Figure 3 the information on the different dilutions is removed and a single homogenized dataset is shown.

new Finally, in Figure 3 the information on the different dilutions is removed and a single dataset per filter, describing the INP concentration as a function of the nucleation temperature is shown.

Lines 105-111: In conclusions, a specific guidance for future research directions is not so clear. Seems to be better explained in the abstract than in conclusions.

We have added a sentence to give some more clear guidance for the future research direction regarding vertical measurements of the INP concentration.

old Since the data is given based on freezing events, the differential spectra can be calculated, obtaining characteristic nucleation temperatures for the aerosol sampled.

new Since the data is given based on freezing events, the differential spectra can be calculated, obtaining characteristic nucleation temperatures for the aerosol sampled. The measurement of the INP concentration at different vertical levels in the lower atmosphere should be extended in the future. Connecting these measurements with ground-based measurements might prove vital in understanding the impact of INPs on weather and climate via primary ice nucleation in mixed-phase clouds.

**References**

Hatakka, J. et al. (2003). "Overview of the atmospheric research activities and results at Pallas GAW station". In: *Boreal Environment Research* 8.

Schmale, J., P. Zieger, and A. M. L. Ekman (2021). "Aerosols in current and future Arctic climate". In: *Nature Climate Change* 11.2, pp. 95–105. DOI: 10.1038/s41558-020-00969-5.